# Psychometric Properties of the Taiwanese Version of the Tilburg Frailty Indicator for Community-Dwelling Older Adults

**DOI:** 10.3390/healthcare9091193

**Published:** 2021-09-10

**Authors:** Chia-Hui Lin, Chieh-Yu Liu, Jiin-Ru Rong

**Affiliations:** 1School of Nursing, National Taipei University of Nursing and Health Sciences, Taipei 112, Taiwan; clh9031@gmail.com; 2School of Long-Term Care and Management, Wufeng University, Chiayi 621303, Taiwan; 3Biostatistical Consultant Laboratory, Department of Health Care Management, National Taipei University of Nursing and Health Sciences, Taipei 112, Taiwan; chiehyu@ntunhs.edu.tw

**Keywords:** frailty, community-dwelling older adults, Taiwanese version of TFI, confirmatory factor analysis, cutoff

## Abstract

Screening the frailty level of older adults is essential to avoid morbidity, prevent falls and disability, and maintain quality of life. The Tilburg Frailty Indicator (TFI) is a self-report instrument developed to assess frailty for community-dwelling older adults. The aim of this study was to explore the psychometric properties of the Taiwanese version of TFI (TFI-T). The sample consisted of 210 elderly participants living in the community. The scale was implemented to conduct a confirmatory factor analysis (CFA) test for validity. The models were evaluated through sensitivity, specificity, area under the curve, and receiving operating characteristic (ROC) curve. CFA was performed to evaluate construct validity, and the TFI-T has a goodness of fit with the three-factor structure of the TFI. Totally, the 15 items of TFI-T have acceptable internal consistency (Cronbach’s alpha = 0.78), and test–retest reliability *(r* = 0.88, *p* < 0.001). The criterion-related validity was examined, the TFI-T correlation with the Kihon Checklist (KCL) score (*r* = 0.74; *p* < 0.001). The cutoff of 5.5 based on the Youden index was considered optimal. The area under the ROC curve analysis indicated that the TFI-T has good accuracy in frailty screening. The TFI-T exhibits good reliability and validity and can be used as a sensitive and accurate instrument, which is highly applicable to screen frailty in Taiwan among older adults.

## 1. Introduction

Frailty is generally associated with the aging process and occurs in 4.0–59.1% of community-dwelling older adults [1]. A study on community frailty in Taiwan found that, among 1014 older adults, 23.1% were diagnosed with pre-frailty and 17.6% were diagnosed with frailty [2]. A systematic review of 46 studies found that after a follow-up of three years, non-frail individuals (from a total of 120,805 participants in 28 countries) became frail with a pooled incidence rate of 43.4% [3]. Several studies have indicated that the prevalence of frailty is increasing in the global aging population [1,3]. Frailty is a clinical syndrome caused by the decline of bodily functions, and it can lead to disability, reduced quality of life, relocation in long-term care institutions, or even death [4,5,6,7,8,9].

Furthermore, frailty has increased reliance on assisted care for the living and associated costs and is a public and community health problem [1,8]. Screening and identifying the frailty conditions in older adults is the major focus of elderly health care, and using a validity tool for health prevention can delay debilitation and disability of older adults.

The accurate assessment of frailty, which can increase frailty detection, moreover, establish validity and an easily assessed frailty instrument, is still a significant clinical and research priority of Taiwan. A systematic review study reported that there is no gold standard for frailty screening, and that existing tools measure different dimensions of frailty, resulting in discrepancies in the reported prevalence in older adults (4–59.1%) [1]. Two measurement tools, the Study of Osteoporotic Fractures (SOF) scale and the Kihon Checklist (KCL), were commonly used for screening frailty in community Taiwan older adults. However, some problems of imprecision or inconvenience exist in operating these tools for assessing and identifying a frailty condition in community-dwelling elderly people. The SOF scale is used for frailty evaluation and fracture prediction in community-dwelling older adults, but it only has three questions (regarding weight loss, reduced lower-extremity function, and reduced energy levels), and is meant to detect frailty at the physical level [10]. However, the SOF scale only contains three items, which likely leads to underestimations of frailty. In addition, a study in South Korea confirmed that the three items of the SOF scale were unable to effectively forecast mortality, function deterioration, and hospitalization [11], and proposed that these inadequacies show the limitations of the scale [10]. On the other hand, the KCL was used for assessing the frailty severity level in Taiwan community-dwelling older adults. The KCL assesses frailty in seven dimensions (25 questions), and can more clearly reveal issues related to frailty and pre-frailty [12,13]; however, comparing with TFI in applying the KCL for older adults is challenging because the number of questions is too many and some of the questions are not a fit for the individual living conditions of community-dwelling older adults in answering questions.

The Tilburg Frailty Indicator (TFI) was developed from an integral understanding of frailty that includes physical, psychological, and social domains [14]. The TFI questionnaire contains 15 questions; it is a user-friendly self-report questionnaire that takes less than 15 min to complete the response. It is also easy to understand and widely administered to community-dwelling older adults [14,15]. Researchers had done a systematic review about TFI as one of the frailty assessment tools which had evidence of reliability and validity within statistically significant parameters and of fair-excellent methodological quality [16]. The original version of the TFI was designed by Gobbens et al. (2010) in the Netherlands; the reliability (Cronbach’s α) and cutoff of the Dutch version of the TFI were 0.73 and 5, respectively [16]. Psychometric tests have since been performed with this instrument, and its diagnostic value, validity, and reliability have been demonstrated. It has been translated from English into several other languages in countries such as China, Germany, Italy, Brazil, Poland, and Portugal [17,18,19,20,21,22]. Although there is a Chinese version of TFI [17], it may not be suitable for populations in Taiwan due to differences in written characters, linguistic expressions, living standards, cultural environment, and education levels. To our knowledge, no similar instrument has been published and validated in the Taiwanese language.

Consequently, the aim of this study was to examine the psychometric properties of a Taiwanese translation of the TFI (TFI-T) in community-dwelling older adults. We also sought to provide initial evidence for its reliability and validity and identified a suitable cutoff point for the TFI-T.

## 2. Materials and Methods

### 2.1. Study Design and Sample

This study adopted an exploratory cross-sectional design. A convenience sampling method was used. In total, 210 older adults living in a community of Taiwan were enrolled in this study. Data on demographic characteristics that include age, gender, marital status, education, monthly income, and measurements of frailty that include SOF, KCL, TFI-T were collected using structured questionnaires from 15 April 2020 to 30 March 2021. Due to the poor eyesight of the older adults, reading and filling out the questionnaires on their own are difficult. Therefore, the researcher read questionnaires and assisted each elderly subject in completing responses to the questionnaires.

The inclusion criteria were age over 60 years and the ability to engage in conscious and coherent verbal communication with the interviewer. The exclusion criteria were a mental disorder diagnosis, drug or alcohol addiction, severe visual or hearing impairment, and refusal to participate.

The sample size was calculated based on each item need to have five times the subjects to calculate the sample size [23]. In this study, the KCL scale has 25 items and that is the most number of items in this study; that is, the sample size needed 125 cases at least. Moreover, we had 210 participants complete all the questionnaires in this study.

### 2.2. TFI

The TFI questionnaire contains two parts: part A, which is used to identify the determinants of frailty, and part B, which is used to identify frailty, which is a standardized self-report questionnaire with 15 items addressing three domains [16]. The three domains measure physical domain (8 items, including physical health, unexplained weight loss, difficulty walking, balance, hearing, vision, hand strength, and fatigue), scored from 0 to 8 points; psychological domain (4 items, including cognition, depressive symptoms, anxiety, and coping), scored from 0 to 4 points; and social domain (3 items, including living alone, social relationships, and social support), scored from 0 to 3 points. The minimum score is 0 and the maximum score is 15 [15]. The psychometric properties of TFI were reported by Gobbens et al. (2010), who showed that a higher score indicates more severe frailty, and a cutoff score of 5 provides a sensitivity of 84% and a specificity of 76% in identifying frailty. The Cronbach’s α for the physical domain was 0.70, the psychological domain was 0.63, the social domain was 0.34, and 0.73 for the total TFI [16]. Significant correlations between the frailty domains were 0.42 between the physical and psychological, 0.19 between the physical and social, and 0.18 between the psychological and social domains (all *p* < 0.001) [14,15].

Gobbens and Uchmanowicz (2021) reviewed 27 studies and found that most of the TFI studies (*n* = 25) were focused on community-dwelling older people, and the internal consistency and test–retest reliability were good. For testing the concurrent validity of the TFI with adverse health outcomes, the following aspects were measured: disability of activities of daily living (ADL) and/or instrumental activities of daily living (IADL), depression (GDS-15), and quality of life (WHOQOL-BREF, WHOQOL-OLD and EUROHIS-QOL). Studies demonstrated that higher scores on the TFI were correlated with lower quality of life [15]. Moreover, regarding disability in performing ADL and/or IADL, the AUCs were acceptable in previous studies [14,19]. One study also reported the relationship between the total domain and psychological domain of TFI and depression (GDS-15), which are 0.67 and 0.49 (*p* < 0.001) [16].

### 2.3. Translation and Cultural Adaptation of the TFI for Taiwan

The repeated forward–backward translation procedure was applied to translate the TFI from English into the Taiwanese language [24]. The original scale was reported in English and then translated into Taiwanese and later reviewed by two bilingual professional translators. It was then translated back into Mandarin for Taiwanese. The original and subsequent versions were compared by two nursing researchers with a master’s degree or Ph.D. degree; minor modifications were made to reach consensus. For content validity, the final version was once again checked. We invited six experts, including physicians, nursing scholars, and public health researchers, to examine the content validity.

Subsequently, a provisional version of the Taiwanese questionnaire was developed, and a pilot study was performed with 30 respondents with older adults. Small revisions have been made to the translated version as a result of the pilot study’s findings. Ultimately, a final Taiwanese version of the TFI was used in this study.

### 2.4. Other Instruments

Besides the TFI, two other frailty instruments were used. The SOF scale is easy to apply, with frailty classified as the presence of two or more components out of three: weight loss, exhaustion, and low mobility. Its categories include frail (score of 2–3), pre-frail (score of 1), and robust (sore of 0) [17]. The KCL consists of 25 questions in the following domains: Activity of daily living (ADL) (Q1–Q5), physical strength (Q6–Q10), nutrition (Q11–Q12), oral function (Q13–Q15), isolation (Q16–Q17), memory (Q18–Q20), and mood (Q21–Q25). Each answer is dichotomous (yes or no), and a point is given for a deficit in each domain. As previously used in validation studies, a score between 0 and 3 is considered robust, KCL cutoff score of 6, ≥7 points indicating general frailty [10,13,25].

### 2.5. Analysis

The continuous variables were displayed using the mean and standard deviation (SD) and the categorical variables were displayed using case number (n) and percentage (%).

After presenting the descriptive statistics, the results of reliability and validity analyses were reported. Both the test–retest reliability and internal consistency reliability of frailty and frailty domain scores in the TFI-T were reported. Test–retest reliability was calculated using the Pearson correlation coefficient. The internal consistency reliability was assessed using Cronbach’s α. The content validity was evaluated using the item content validity index (I-CVI) and the scale-level content validity index (S-CVI).

Additionally, the SOF scale and KCL were used to test the criterion-related validity of the TFI-T. The receiver operating characteristic (ROC) curve analysis was applied to assess the criterion-related validity of the TFI-T with other specific frailty measures (i.e., the KCL and SOF). The diagnostic index (DI: specificity + sensitivity) and Youden’s index [γ = sensitivity − (1 − specificity)] were calculated as a reference for the suitability of the cutoff point after the sensitivity and specificity were calculated using the area under the ROC curve (AUROC). Construct validity of frailty was also assessed using convergent validity Pearson’s correlation coefficient. We adopted the TFI-T cutoff score into two groups (frail and robust), and then performed independent *t*-tests on the total SOF scale, total KCL scores, TFI-physical, psychological and social domain to examine discriminant validity. Confirmatory factor analysis (CFA) was used to test the goodness of fit of the conceptual framework of the TFI-T.

Statistical analysis was conducted using SPSS v26.0 (IBM Corporation, Armonk, NY, USA) [26], and CFA was conducted using LISREL v 8.8 [27]. A two-tailed significance level of 0.05 was considered statistically significant.

### 2.6. Ethical Considerations

This study was approved by the Jianan Psychiatric Center at the Ministry of Health and Welfare in Tainan city (IRB Approval Number: 02-012). Moreover, the researcher provided written information for explaining to the participants the research purpose, data collection process, and protection of individual rights, in terms of participation, anonymity, and confidentiality. Participants in the research are voluntary and free to withdraw from the research at any time. After the participants agreed to participate in this study, they were requested to fill out the informed consent form and the questionnaires, the researcher read questionnaires and assisted each elderly subject in completing responses to the questionnaires, which took approximately 30–40 min each to complete.

## 3. Results

### 3.1. Participant Characteristics

At the end of the study period (30 March 2021), 210 community-dwelling older adults had been contacted. Table 1 presents the participants’ characteristics. Their ages ranged from 60 to 94 years, and the mean age was 75.45 years (SD = 9.15). Most of the participants were female (75.7%). Nearly half of the participants were married or widowed. A majority had received more than 10 years of education (51%), and more participants had some income (86%). The mean TFI-T total score, SOF total index and KCL total score were 5.69 (SD = 3.22), 0.97 (SD = 1.05) and 7.27 (SD = 4.49), respectively.

### 3.2. Reliability and Validity

#### 3.2.1. Internal Consistency Reliability

Cronbach’s α coefficients of the TFI-T were 0.78 for the total scale, 0.79 for the physical domain, 0.79 for the psychological domain, and 0.81 for the social domain, indicating good internal consistency [28]. These values were similar to those reported in the original studies (Cronbach’s α values were 0.34, 0.63, 0.70 respectively) [15,16].

#### 3.2.2. Test–Retest Reliability

The study tested the 15-item questionnaire twice to 30 community-dwelling older adults with a three-week time interval. The mean total score on the first test was 5.45 (SD = 3.27), and the mean total score on the second test was 5.36 (SD = 3.39); these results indicate that the test and retest did not significantly differ, yielding a test–retest coefficient of 0.88 (*p* < 0.001). Therefore, test–retest reliability was adequate.

#### 3.2.3. Content Validity

We invited six experts, including physicians, nursing scholars, and public health researchers, to examine the content validity. The experts rated most of the items in the TFI-T as highly relevant, leading to an acceptable average. The I-CVI values ranged from 0.9 to 1.0, and the S-CVI was 0.9.

#### 3.2.4. Criterion-Related Validity

We assessed the criterion-related validity of TFI -T by calculating correlation coefficients among the SOF and KCL. The criterion-related validity analysis was conducted using Pearson’s correlation coefficient. The TFI-T total score was strongly and significantly correlated with the KCL score (Pearson’s *r* = 0.742; *p* < 0.001), but SOF scale was not significantly correlated with the TFI-T (Pearson’s r = 0.11; *p* > 0.001) and KCL score (Pearson’s *r* = 0.13; *p* > 0.001) (Table 2). In addition, the correlation between the TFI-T total score and fifteen items of the TFI-T were all statistically significant (Pearson’s *r* = 0.26–0.70, *p* < 0.01).

Concurrent validity was also examined using ROC curve analyses. With the KCL as the criterion for diagnosing frailty, the AUROC of the TFI-T and SOF scale was 0.87 (95% CI: 0.79–0.90) and 0.55 (95% CI: 0.52–0.70), respectively (Figure 1). The ROC curve graphically displays the trade-off between sensitivity and specificity and is useful in assigning the best cutoffs for clinical use. The results of our determination of the optimal cutoff point for the TFI-T, based on the Youden index, showed an optimal cutoff of 5.5, sensitivities of 76.4%, and specificities of 83% (Table 3).

We had the dilemma created by the trade-off between sensitivity and specificity (Table 3). A cut-off point on the TFI-T of 5.5 will miss only 23.6% of frail adults, but 17% of robust adults by a false-positive report, and the DI is 1.59. Raising the cutoff to 6.5 reduces false-positive reports to 9% of the non-frail adults, at the expense of missing nearly 35% of the frail older adults, and the DI is 1.56. However, the sensitivity of the score is more important in a clinical setting; however, a cutoff score of 5.5 is recommended.

#### 3.2.5. Construct Validity

In the results for the final CFA model, the three-factor model demonstrated satisfactory model fit indices (χ2 = 174.3; *df* = 87; GFI = 0.98; CFI = 0.91; AGFI = 0.83; RMR = 0.04; NFI = 0.84; RMSEA = 0.0054; NFI = 0.89) (Table 4). The estimated GFI, CFI, and NFI are 0.90 or greater and the RMR and RMSEA are 0.05 or less [28], it shows that the model does not violate the estimation, which means the best model adopted in this research is fit for re-search. The standardized item-loading model ranged from 0.20 to 0.87. Although the loading of three items (Q2, Q14, Q15) were lower than 0.3, however, in order to stay consistent with prior studies and provide a better diagnostic criterion, Q2, Q14, Q15 were retained in the present scale (Figure 2). In general, considering the content integrity and the consistency with the prior study, the three domains with 15 items were accepted.

As shown in Table 5, each domain of TFI-T significantly correlated with KCL domains as expected. The convergent validity of the TFI was affirmed by the Pearson’s coefficient between each item of the TFI-T and KCL domains. All of the *r* values ranging from 0.14 to 0.61 were statistically significant (Table 5).

We adopted the TFI-T cutoff score of 5.5 and divided the 210 respondents into two groups: frail (TFI-T ≥ 5.5) and robust (TFI-T < 5.5). We then performed independent *t*-tests on the total SOF scale, total KCL scores, TFI-physical, psychological and social domain. The results indicated significant statistical differences between the KCL score of the frail group and that of the robust group (*t* = −11.29; *p* < 0.001). Referencing a KCL cutoff score of 6, ≥7 points indicating general frailty [14], the corresponding total KCL score of the frail group was 10.17 ± 4.12. Similarly, the three TFI-T subdomains also showed significance between the two groups. These results indicate that the TFI-T and KCL could be used to effectively identify frailty, demonstrating discriminant validity (Table 6).

## 4. Discussion

### 4.1. General Discussion

The aim of this study was to translate the TFI to the Taiwanese version and to assess the questionnaire’s psychometric properties in Taiwan community-dwelling older adults. Frailty is a dynamic condition, which includes physical frailty, social frailty, psychological frailty, and total frailty [16]. An advantage of this tool includes its user-friendly nature as a questionnaire that can be completed efficiently in 15 min or less without a direct interview [14,15].

In terms of the psychometric value of the TFI-T, the reliability was acceptable with internal consistency (Cronbach’s α) of 0.78 [28]. The value was close to that reported in previous studies (0.71–0.78) [14]. In this present study, the test–retest reliability coefficient of 0.88 was good, with similar results as previous studies (0.79–0.80) [14,17].

This study examined the construct validity of using CFA, convergent validity, and discriminant validity. The results of CFA were accepted and demonstrated the TFI-T can fit the structure of TFI, with three factors include 15 items. However, three items of TFI-T showed low factor loading in the instrument structure. In order to explore whether deleting these three items would affect the quality of the scale, this study calculated the Cronbach’s α after deleting these items. The total TFI-T Cronbach’s α changed from 0.780 to 0.789 after deletion. The change in coefficient is very slight and considered the differences related to the sample composition, culture and other factors. There was no need to exclude any items of TFI-T. Moreover, the convergent validity was tested and the TFI-T correlates significantly with the KCL, and three factors of TFI-T strongly and significantly correlated with the domains of KCL with its corresponding frailty measure.

The TFI-T exhibits good reliability and validity and can be used as a sensitive and accurate instrument, which is highly applicable to screen frailty in Taiwan among older adults. Using the KCL as a reference criterion, the AUC of the TFI-T showed good diagnostic accuracy in the identification of frailty. Mandrekar (2010) stated that an AUC of 0.8 to 0.9 suggests excellent discrimination (i.e., ability to diagnose patients with and without the disease or condition based on the test) [29]. In this study, the results of the AUC analysis indicate that the TFI-T for predicting frailty (AUC: TFI = 0.87, *p* < 0.001), that is, the TFI-T has good accuracy and excellent discriminating in assessing frailty in Taiwanese older adults. In addition, This study determined the optimal cutoff and corresponding diagnostic accuracy of the TFI-T for the frailty screening of older adults in Taiwan. Based on the Youden index, a cutoff of 5.5 was considered optimal. The reported cutoff value is similar to those established for the Netherlands and Polish versions (both version cutoff values are 5.0) [15,21].

### 4.2. Limitations

This study has some limitations that need to be mentioned. First, it is a cross-sectional study that cannot interfere with the causal relationship between frailty and adverse health outcomes, which means that this study may not provide the predictive validity of the TFI-T [30]. Second, the data collection of the TFI-T scale in this study was only completed by the first author; therefore, when TFI-T is used to measure the frailty assessment of the elderly people in the community, the assessment may be performed by different data collectors. The consistency of the data collection between data collectors needs to be re-evaluated [31]. In the future, when promoting community health professionals to use the TFI-T scale to assess the frailty of the elderly in the community, instrument training and consistency testing among data collectors are required. Third, the instrument validity and cutoff value analysis need tests with the physical measures or medical examination for exploring the individual problems and their care needs to manage the frailty. Further longitudinal studies and investigations of the frailty are recommended to more effectively explain the changes in and cutoff point scores. It is recommended that future studies examine the differences of frailty among sub-groups of sample characteristics and inferential the relationship between the frailty and related health outcomes.

## 5. Conclusions

This study translated the TFI for use among community-dwelling older adults in Taiwan. The TFI-T exhibits good reliability and validity and can be used as a sensitive and accurate instrument, which is highly applicable to screen frailty in Taiwan among older adults.

## Figures and Tables

**Figure 1 healthcare-09-01193-f001:**
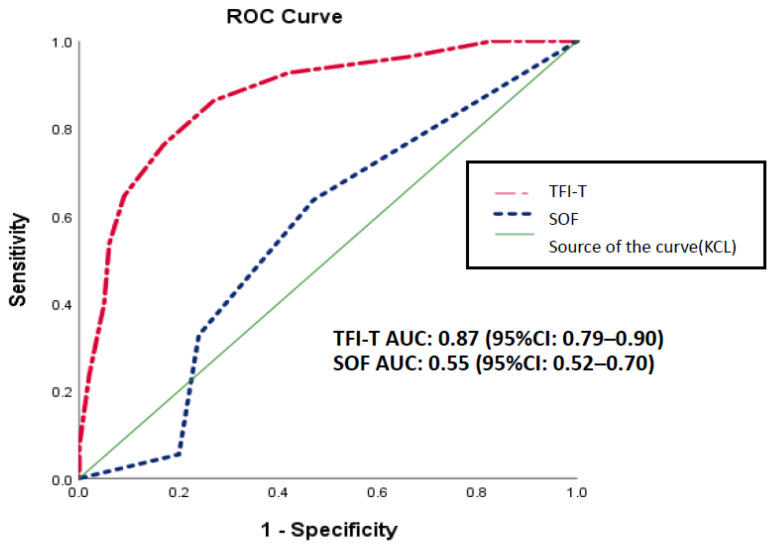
ROC curve of the TFI-T for establishing a cutoff score for frailty.

**Figure 2 healthcare-09-01193-f002:**
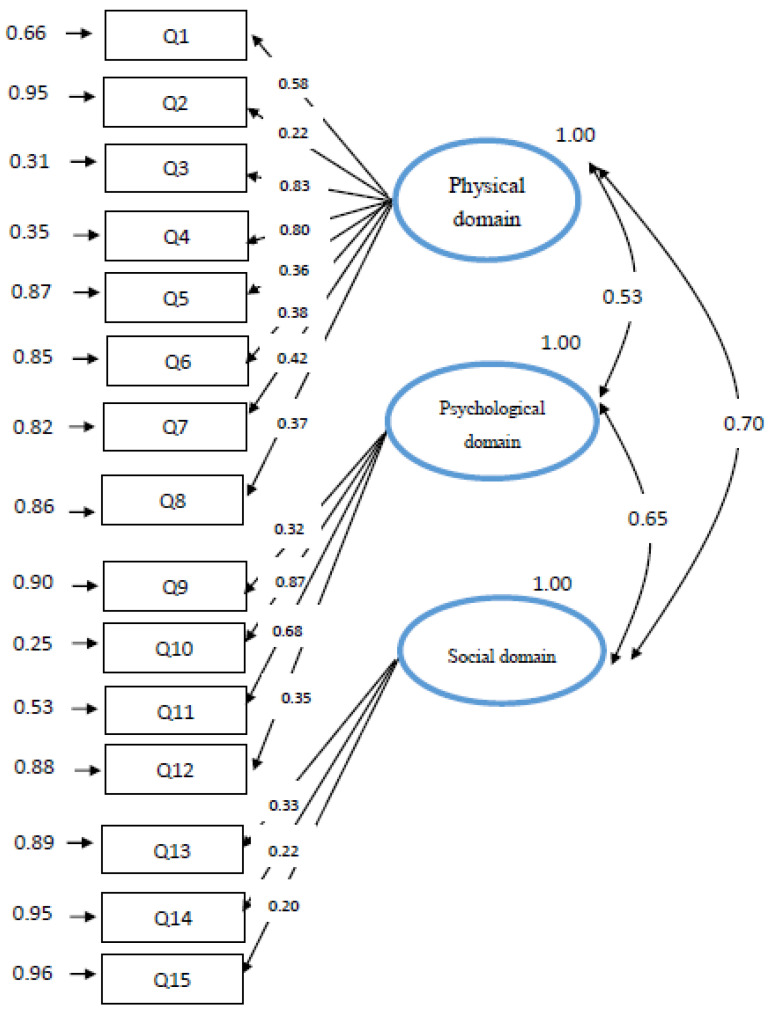
Final CFA model the TFI-T.

**Table 1 healthcare-09-01193-t001:** Characteristics of participants (*N* = 210).

**Variable**	** *n* ** **(%)**
Age (years)	
60–69	61 (29.0%)
70–79	68 (32.4%)
80–94	81 (38.6%)
Gender (female)	159 (75.7%)
Marital status	
Married or remarried 100	(48%)
Widowed	97 (46%)
Divorced and unmarried	
≤6	53 (25%)
7–9	54 (26%)
≥10	103 (49%)
Monthly income	(Taiwan Dollars)
No income 29	(14%)
<10,000	100 (47%)
10,001–50,000	81 (39%)
**Variable**	**Mean ± SD**
SOF total index (0–3)	0.97 ± 1.05
KCL total score (0–25)	7.27 ± 4.49
KCL sub-domain:	
ADL	1.10 ± 1.49
physical strength	1.87 ± 1.48
Nutrition	0.25 ± 0.48
oral function	1.24 ± 0.92
Isolation	0.34 ± 0.55
Memory	1.23 ± 0.88
Mood	1.25 ± 1.36
TFI-T total score (0–15)	5.69 ± 3.22
TFI-T sub-domain:	
Physical domain score (0–8)	3.19 ± 2.18
Psychological domain score (0–4)	1.18 ± 1.19
Social domain score (0–3)	1.32 ± 0.64

**Table 2 healthcare-09-01193-t002:** Correlation matrix for the TFI-T, SOF and KCL (*N* = 210).

Variable	1	2	3
1. SOF	1	-	-
2. KCL	0.13	1	-
3. TFI-T	0.11	0.74 ***	1

*** Correlation is significant at the 0.001 level (two-tailed).

**Table 3 healthcare-09-01193-t003:** The best cutoff point for the TFI-T based on Youden’s index.

Criterion	Cutoff	Sensitivity (True Positive Rate) (%)	Specificity (True Negative Rate) (%)	DI	Youden’s Index
KCL	TFI-T ≥ 4.5	86.4	73	1.59	0.59
	TFI-T ≥ 5.5	76.4	83	1.59	0.59
	TFI-T ≥ 6.5	65.0	91	1.56	0.56
	TFI-T ≥ 7.5	53.6	94	1.47	0.49

**Table 4 healthcare-09-01193-t004:** Construct validity of the TFI-T.

Three-Factor Model	X^2^/*df*	GFI	CFI	AGFI	RMR	NFI	RMSEA	NFI
Scale	174.3/87	0.98	0.91	0.83	0.04	0.84	0.0054	0.89

X^2^ = Chi-square, df = degrees of freedom. GFI = Goodness-of-fit index, CFI = Comparative fit index, AGFI = Adjusted goodness of fit index, RMR= Regional mobile radio, RMSEA= root mean square error of approximation, NFI = Normed fit index.

**Table 5 healthcare-09-01193-t005:** Correlation between the TFI-T domains and KCL domains.

Domains of TFI-T	Domains of KCL	*r*	*p*-Value
Physical domain (Q1–Q8)	ADL (Q1–Q5)	0.43	<0.01
	physical strength (Q6–Q10)	0.61	<0.01
	Nutrition (Q11–Q12)	0.14	<0.05
	oral function (Q13–Q15)	0.37	<0.01
Psychological domain (Q9–Q11)	Memory (Q18–Q20)	0.39	<0.01
	Mood (Q21–Q25)	0.45	<0.01
Social domain (Q12–Q15)	Isolation (Q16–Q17)	0.30	<0.01

**Table 6 healthcare-09-01193-t006:** Analysis of the difference between the two groups of TFI-T (*N* = 210).

Variable	TFI-T		*t*	*p-*Value
	<5.5 robust group (*n* = 109) (mean ± SD)	≥5.5 frail group (*n* = 101) (mean ± SD)		
SOF scale	0.85 ± 1.15	1.09 ± 0.91	−1.63	0.10
KCL score	4.59 ± 2.86	10.17 ± 4.12	−11.29	<0.001
TFI-T Physical domain	1.49 ± 1.05	5.02 ± 1.49	−19.60	<0.001
Psychological domain	0.44 ± 1.97	1.97 ± 1.12	−11.77	<0.001
Social domain	1.14 ± 0.56	1.52 ± 0.65	−4.54	<0.001

## Data Availability

For original data, please contact the corresponding author; ethical approval does not cover making data openly accessible.

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
