# Peer review of "Psychometric Properties of the Taiwanese Version of the Tilburg Frailty Indicator for Community-Dwelling Older Adults"

_healthcare, 2021, doi:10.3390/healthcare9091193_

Round 1
Reviewer 1 Report
Dear authors,
I would like to congratulate you on your work. This study shows the results of the adaptation to the Taiwanese language of the TFI scale. Robust and well-performed, no needs an in-deep consideration except a few remarks.
Page 2 line 17, please check the typo before [10] citation reference.
Page 2 line 23, please check the space before the parenthesis.
Page 4: cite SPSS software as IBM Corp. Released 2019. IBM SPSS Statistics for Windows, Version 26.0. Armonk, NY: IBM Corp and LISREL as an adjunct document.

Author Response
Comments and Suggestions for Authors |
Authors responses |
1.Page 2 line 17, please check the typo before [10] citation reference. |
Thank you for your suggestion. We have reflected this comment by fixed typo (p. 2, lines16). Please see the yellow mark. |
2.Page 2 line 23, please check the space before the parenthesis. |
Thank you for your suggestion. We have reflected this comment by delete the space (p. 2, lines 22). Please see the yellow mark. |
3.Page 4: cite SPSS software as IBM Corp. Released 2019. IBM SPSS Statistics for Windows, Version 26.0. Armonk, NY: IBM Corp and LISREL as an adjunct document. |
Thank you for your suggestion. The citation of IBM SPSS has been revised at p. 4, line 34-36. |

Reviewer 2 Report
I want to congratulate the authors on the subject of the manuscript. In my opinion, I found the study very interesting and I think the topic is very necessary. The manuscript is written in an understandable way and contains in each section the most relevant aspects of the research.
Abstract:
Background and use of appropriate abbreviations. The type of study is clear.
The title: It is adequate, concise and clear. The descriptors (keywords) of the study are clearly identified.
- Introduction:
The references in the text, justifying data and the international scope are correct.
- Materials and methods:
-The study design and type of sampling criteria for inclusion, exclusion, losses, ... are well defined.
- The sample could have been stratified by age, sex and size of the place of residence.
2.1 Study design and sample.
It would be necessary to specify how the interview was carried out, context, duration, information sheet, ... In general, there is a lack of information on data collection, procedures and measurement of the different types of variables.
Ethical Considerations: Confidentiality and anonymity are not mentioned in the collection, analysis and custody of data. The variables are not clear enough.
The size of the sample calculation is correctly defined.
-Appropriate data collection: instruments, validity.
-For the validity of the criteria, the scores of the Tilburg Indicator (total score and by domains) could be compared with other scales that measure aspects of frailty (WHOQL and Yesavage's Test).
- Results
Correct presentation and design of the results, adequate use of figures and tables with different designs. Sometimes the facts and figures of the text are repeated in the table.
- Discussion
-The order in the presentation of the discussion is appropriate.
-No reference is made to the limitations of the study, type of residences (socioeconomic context) that live in the Taiwanese community, health personnel, services provided, type of management, material and human resources according to the different areas of the community. Who and how the interview process is carried out, rules, concordances.
Better define the homogeneity of the sample, 75% women, ...
Appropriate, current and international references.
Author Response
Comments and Suggestions for Authors |
Authors responses |
Abstract: Background and use of appropriate abbreviations. The type of study is clear. The title: It is adequate, concise and clear. The descriptors (keywords) of the study are clearly identified. |
Thank you for your comment. |
Introduction: The references in the text, justifying data and the international scope are correct. |
Thank you for your comment. |
Materials and methods: -The study design and type of sampling criteria for inclusion, exclusion, losses, ... are well defined. - The sample could have been stratified by age, sex and size of the place of residence. |
1.Thank you for your comment. 2.Stratified sampling uses specific characteristics; it can provide a representation of the population based on what's used to divide it into different subsets. In this study, the elderly people are a major sample, age group of the sample includes young-old (60-69 years, 29%), middle-old (70-79 years, 32.4%), and oldest-old age (80-94 years, 38.6%) group (see Table 1). A convenience sampling was used in this study, and the old age subgroup can provide sufficient data points to represent the community elderly population for test the Taiwanese version of TFI. |
2.1 Study design and sample. |
|
(1) It would be necessary to specify how the interview was carried out, context, duration, information sheet, ... In general, there is a lack of information on data collection, procedures and measurement of the different types of variables. |
1.Thank you for your suggestion. 2. We have read your suggestion and amended it in the text (p. 2, lines52; p.3, lines 1-5) Please see the yellow mark. |
(2) Ethical Considerations: Confidentiality and anonymity are not mentioned in the collection, analysis and custody of data. The variables are not clear enough. |
1.Thank you for your suggestion. 2. We have read your suggestion and amended it in the text (p. 4, line 39-46.) |
The size of the sample calculation is correctly defined. -Appropriate data collection: instruments, validity. |
Thank you for your comment. |
(3)For the validity of the criteria, the scores of the Tilburg Indicator (total score and by domains) could be compared with other scales that measure aspects of frailty (WHOQL and Yesavage's Test). |
1.Thank you for your suggestion. 2.We have read your suggestion and amended it in the text (p. 3, line 31-40.) Please see the yellow mark. |
Results Correct presentation and design of the results, adequate use of figures and tables with different designs. Sometimes the facts and figures of the text are repeated in the table. |
Thank you for your suggestion. To facilitate reading, only a few repetitive figure and text descriptions are kept. |
Discussion -The order in the presentation of the discussion is appropriate. |
Thank you for your comment. |
-No reference is made to the limitations of the study, type of residences (socioeconomic context) that live in the Taiwanese community, health personnel, services provided, type of management, material and human resources according to the different areas of the community. Who and how the interview process is carried out, rules, concordances. Better define the homogeneity of the sample, 75% women, ...Appropriate, current and international references. |
1.Thank you for your suggestion. 2.We have read your suggestion and amended it in the text (p. 9, line 38-51. And p.10, line 1-2). Please see the yellow mark.
|
